# Evolution of Molecular Biomarkers and Precision Molecular Therapeutic Strategies in Glioblastoma

**DOI:** 10.3390/cancers16213635

**Published:** 2024-10-29

**Authors:** Maria A. Jacome, Qiong Wu, Yolanda Piña, Arnold B. Etame

**Affiliations:** 1Departamento de Ciencias Morfológicas Microscópicas, Universidad de Carabobo, Valencia 02001, Venezuela; 2Department of Neuro-Oncology, H. Lee Moffitt Cancer Center and Research Institute, 12902 Magnolia Drive, Tampa, FL 33612, USA; qiong.wu@moffitt.org (Q.W.); yolanda.pina@moffitt.org (Y.P.)

**Keywords:** glioblastoma, molecular biomarkers, molecular targeted therapy, gene therapy

## Abstract

Glioblastoma is an extremely lethal malignant brain tumor. Finding ways to improve current treatments and outcomes for patients is crucial. Molecular profiling has become essential in diagnosis and management, with new technologies in areas of histopathology and radiogenomics being currently developed. Molecular biomarkers are the target of new therapies that hold great potential for refined and personalized treatments that aim to improve patient survival. This review summarizes the latest advances in the fields of histopathology and radiogenomics and the development of targeted therapies, providing an overview of the results of recent trials and the future directions of molecular targeted therapies in glioblastoma.

## 1. Introduction

Worldwide, CNS tumors represent roughly 1.7% of all diagnosed cancers [1]. Glioblastoma, also referred to as grade 4 IDH wild-type astrocytoma, is the most common malignant brain tumor, accounting for 14.2% of all CNS tumors, and has a 5-year survival rate of only 6.9% [2]. Despite being a relatively rare disease, the high mortality rate of glioblastoma has ignited the field of investigation with the goal of finding better diagnostic tools and treatment options for glioblastoma.

Glioblastoma was the first cancer type whose genetic landscape was extensively characterized by The Cancer Genome Atlas Research Network (TCGA) in its initial publication in 2008 [3]. Subsequently, as part of the revision of its fourth edition in 2016, the World Health Organization Classification of Tumors of the Central Nervous System (WHO CNS4) began incorporating molecular biomarkers as an essential tool for characterization of tumors for greater diagnostic accuracy, prognosis and patient management [4]. Most recently, the fifth edition of the WHO Classification (WHO CNS5) published in 2021 established molecular and genetic features as fundamental for CNS tumor diagnosis and subsequent management [5,6]. Diagnostics nomenclature is standardized to include histopathology, CNS WHO grading and molecular findings [7]. With molecular refinements in glioblastoma diagnosis and oncogenesis, there has been renewed emphasis on molecularly directed precision medicine approaches.

In this review, we focus specifically on the evolution of molecular biomarkers of glioblastoma, as well as their correlation with prognosis, radiographic presentation, pathological features and individualized responses to different treatment modalities and protocols.

## 2. Current Nomenclature for Classification of Tumors

Historically, histopathology has been the gold standard for classification of CNS tumors, including gliomas. However, recent advances in tumor molecular genetic testing have resulted in paradigm shifts in prognostic classifications of gliomas centered on isocitrate dehydrogenase (IDH) status. IDH mutational status segregates gliomas into favorable prognostic IDH mutants and IDH wild types that portend a poorer prognosis, regardless of histological features [8]. This was intensely discussed by the Consortium to Inform Molecular and Practical Approaches to CNS Tumor Taxonomy (cIMPACT-NOW) [9,10,11] until the WHO CNS5 definitively incorporated an integrated diagnosis approach using histological and molecular information to classify gliomas [6,12,13]. Glioblastoma, IDH-wildtype grade 4 astrocytoma is included as one of the three types of adult-type diffuse glioma listed in the WHO CNS5 classification, with the other two being astrocytoma with IDH-mutant and oligodendroglioma with IDH-mutant and 1p/19q-codeleted [6].

Grading, on the other hand, is no longer based in the conjunction of factors such as mitotic activity and histological features but is mainly based on the tumor’s expected natural history, which is known to be more accurately determined by molecular markers [6]. In that way, glioblastoma IDH-wildtype is graded as a WHO grade 4, given its grim overall prognosis, in disregard of any other specific histologic, radiographic or molecular feature.

## 3. Histopathological Features of Glioblastoma

The histological classification of tumors of glial origin has traditionally been based on the cell of origin, given that oligodendrocytes produce oligodendrogliomas, astrocytes produce astrocytomas and so forth [14]. However, the cell of origin of glioblastoma remains an issue of debate, with recent mounting evidence that glioblastoma originates from neural stem cells (NSCs), especially those from the subventricular zone (SVZ) in the adult brain [15,16]. SVZ contains the brain’s largest population of NSCs, which are multipotent cells that can differentiate into neurons, astrocytes and oligodendrocytes, even in adults [16,17].

The cellularity of glioblastomas supports the hypothesis of NSCs being their cells of origin, given the presence of cells expressing stem cell surface markers within the tumor, combined with the overall complex cellular composition of its core [14,18]. There is evidence that glioblastoma cells actually can include four different cellular states, including (1) neural progenitor-like (NPC-like), (2) oligodendrocyte progenitor-like (OPC-like), (3) astrocyte-like (AC-like) and (4) mesenchymal-like (MES-like) cells [19]. All of these have been proposed as potential cells of origin of glioblastoma [18,20], but no definitive conclusion has been reached. Multi-region tumor sampling has demonstrated that cellularity can even vary and co-exist across different regions of the same tumor, making it harder to determine a single cell of origin [19].

The use of machine deep learning algorithms to diagnose glioblastoma and predict the main cell type based on histology sections alone has been in development in recent years, which could allow for the integration of molecular characteristics and histological features in a cheaper and more readily available way [21,22,23].

There are certain histological and immunohistochemical features that are classic and that, for many years, helped distinguish this tumor from other diffuse gliomas. Local tissue invasion, especially along deep white matter tracts; multiple mitotic figures; nuclear atypia; high nuclear pleomorphism; nuclei with clumped chromatin; microvascular proliferation; and pseudopallisading necrosis (also called Scherer structures) are all characteristic features [4,12,14,24]. The latter two particularly determine the grading of glioblastoma as a WHO grade 4 diffuse glioma and are still considered as valuable as molecular features to confirm a glioblastoma diagnosis in an IDH-wildtype diffuse glioma [6]. Therefore, thee integrated current diagnosis of glioblastoma IDH-wildtype shows a mix between histopathological and molecular features, as shown in Table 1.

Thrombosis has been proposed as a diagnostic criterion for glioblastoma shown to independently predict wildtype IDH status. The concept behind it is the relationship between thrombosis and necrosis, which is stronger in glioblastoma IDH-wildtype, perhaps due to the intrinsic antithrombotic activity of mutant IDH-1 [14,25,26]; however, there is not enough evidence to support its widespread use for diagnosis.

Among the limitations of histopathological examination is the high level of intratumoral heterogeneity of glioblastoma, leading to intraobserver and interobserver variabilities that negatively impact specificity for diagnosis and prognostic utility, since even histologically similar tumors can have different disease progression and outcomes [27,28,29]. There have been a couple of ways to overcome these issues. The first approach entails adequate neurosurgical samplings from multiple regions of the tumor to accommodate variabilities [14]. The other approach involves incorporation of immunohistochemical tests like the Ki-67/MIB-1 test to evaluate proliferative activity as a surrogate for the mitotic activity of tumors, which correlates with higher histological grade and poorer prognosis [30,31].

Still, Ki-67/MIB-1 is not an infallible test, and it should always be used in combination with other histopathological and molecular features of malignancy. Pathologists can encounter great variation of Ki-67 that could range between 5 and 70% from different regions of the same tumor [14]. There is great variability when comparing glioblastoma diagnosed only by histological features, usually with higher Ki-67 values, versus glioblastomas diagnosed molecularly, with statistically significant lower values [12]. It is now apparent that molecular changes can be detected even before high rates of mitotic activity and histological changes develop, hinting at the superior utility of molecular markers over both histological and immunohistochemistry analysis of tissue. Inherent limitations of Ki-67/MIB-1 immunohistochemistry also include sampling errors, leading to difficulties characterizing immunoreactive tumor cell nuclei [30], and the lack of statistically significant differences in survival between cutout values of Ki-67 [32].

Other immunohistochemistry markers usually considered and analyzed for glioblastoma are Glial Fibrillary Axonal Protein (GFAP), S-100, vimentin, oligodendrocyte transcription factor 2 (Olig2) and ATRX, with some finding at least one positive marker in more than 90% of glioblastomas [24,30]. But despite studies conducted to determine the utility of these markers for prognosis or grading, the evidence is not conclusive, and they are not currently used as determinants for diagnosis or prognosis in glioblastoma tumors [31,33].

The workup for glioblastoma is no longer based on one particular aspect but on multiple components. Within this paradigm, contemporary pathology workflows rely on an integrated approach in which traditional histopathologic features are examined in tandem with ancillary molecular tests. Furthermore, breakthroughs in digital pathology allowing for digitized slides known as high-resolution whole-slide images (WSIs) to be also analyzed by computational tools and deep convolutional neural networks (CNNs) have facilitated more precise results. However, this is still a small field with plenty of opportunity for investigation [34].

## 4. Radiographic Presentation of Glioblastoma

### 4.1. Criteria for Assessment of Imaging in Brain Tumors

For decades now, imaging has been the cornerstone of diagnosis of brain tumors. The classical appearance and radiographic behavior of glioblastoma have been well described in standard-of-care MRI sequences. It typically shows an intraparenchymal mass with a heterogeneous, irregularly enhanced signal [35]. But variations can occur, and atypical imaging findings could be confounding and ultimately require histopathological analysis for diagnosis [36]. Differentiation of vasogenic edema, radiation-induced gliosis and real infiltrating tumors often requires more than one MRI sequence and specific criteria for assessment of these changes, especially after treatments [37].

There are consensus recommendations for a standardized brain tumor MRI protocol that includes the following sequences: a precontrast T1, a postcontrast T1 that matches parameters to precontrast T1, an axial T2/FLAIR, axial diffusion-weighted imaging (DWI), and an axial T2, all performed on a minimum 1.5 Tesla—and ideally a 3 Tesla—MR system [38,39], leaving advanced techniques such as DWI-derived apparent diffusion coefficient (ADC) imaging, diffusion tensor imaging (DTI), susceptibility-weighted imaging (SWI), magnetic resonance spectroscopy (MRS) and dynamic contrast-enhanced (DCE) perfusion imaging mainly for use in research studies. The use of these advanced MRI techniques is limited by the need for dedicated software not widely available in all centers, the long time it takes to perform them and the fact that the information they end up providing has not yet been enough to validate a change in patient management [40,41].

The Response Assessment in Neuro Oncology (RANO) criteria, first presented in 2010 specifically for high-grade gliomas [42], offers an upgrade to the previous Macdonald criteria [43]. It integrates the use of steroids and neurologic findings in the assessment of the efficacy of therapies while accounting for radiographic T2/FLAIR changes and not only areas of enhancement in T1 to define tumor progression and response to therapy. It also provides guidance regarding the definition of measurable versus non-measurable disease, multiple-lesion measurements and differentiation between true progression and pseudoprogression in the first 12 weeks after completion of radiotherapy, in addition to considering the previous use of antiangiogenic therapies such as Vascular Endothelial Growth Factor (VEGF) inhibitors [42].

### 4.2. Standard and Advanced MRI Imaging of Brain Tumors

Neuroimaging continues to evolve as a field, and studies keep trying to assess intrinsic tumor features like metabolism, microenvironment and molecular profile using advanced MRI techniques. For example, in a small set of patients, Kamimura et al. [44] proved that changes in ADC values in the enhancing regions of brain tumors may be useful in differentiating brain metastases from glioblastomas, theorizing that either cell size or histological characteristics like cell–cell adhesion may cause the difference in water diffusion seen in imaging [44]. Other studies have aimed to prove that ADC measurements can predict IDH molecular profiling, proposing that IDH-wildtype gliomas have lower ADC values than IDH-mutant tumors [45,46,47]. However, ADC measurements are susceptible to artifacts, especially after treatments that modify vascular permeability, and there is high variability dependent on image quality and tumor enhancement [40,48].

MRS, a water-suppressed proton (^1^H) technique typically employed in brain tumors to assess the concentrations of creatine, choline and n-acetylaspartate (NAA), among other metabolites, has been studied to detect 2-hydroxyglutarate (2-HG) in low-grade gliomas [40]. 2-HG is catalyzed from α-ketoglutarate by a mutant IDH1 such as the one found in grade 2 and 3 gliomas and is, therefore, termed an “oncometabolite” [49]. Detection of 2-HG with edited MRS sequences has reached 100% specificity and 100% sensitivity in some series [50] and been proven to be an accurate noninvasive tool to diagnose and grade gliomas [51]. However, glioblastoma, by definition, lacks an IDH mutation; therefore, measurement of 2-HG is of no utility, which has shifted the investigation towards other metabolites, such as different concentration ratios of choline/NAA and choline/creatine to differentiate true progression of glioblastoma from pseudoprogression [52]. Long scan times, extensive processing and the need for expert operators decrease comparability between studies and represent some of the limitations with respect to the routine use of MRS, which continues to be considered experimental, as well as other techniques, such as sodium MRI, amine-weighted CEST (chemical exchange saturation transfer) and APT (amide proton transfer) CEST [40,53,54].

However, not all research has yielded favorable results. Izquierdo et al. tried to characterize a series of IDH-wildtype, TERT-promoter mutant tumors with MRI, but results showed no radiological association with histological grade, EGFR amplification, MGMT methylation, or chromosome 7 gain and chromosome 10 loss [55], and even the radiological features they observed, like tumor infiltration, enhancement and localization, are difficult to extrapolate to bigger cohorts.

### 4.3. Radiogenomics in Tumors of the Brain

Radiogenomics, also known as imaging genomics, is the specific mining of radiomic data to correlate genetic alterations and imaging features, providing noninvasive and global assessment of tumors—or a “virtual biopsy” [56,57,58]. However, managing and processing all these data is time-consuming, and newer approaches to radiogenomics are pointing in the direction of the use of artificial intelligence (AI), machine learning and deep learning-based models. The application of deep learning using CNNs and AI-based image postprocessing algorithms can potentially improve noninvasive diagnosis, especially when combined with advanced MRI techniques [57,59].

CNNs have allowed radiomics analysis to assess images beyond what is perceivable by the naked eye, and some models have automated the processes in ways that are independent of operators, making them highly efficient [60]. For example, one fundamental step in radiomics analysis is tumor segmentation, which allows for 3D volumetric analysis of tumors [40,58]. However, manual delineation is tedious, time-consuming and vastly dependent on operator expertise [61,62]. Advances in semiautomated and automated computational methods with the use of AI and CNNs have allowed for more efficient segmentation, and with every new study and training algorithm, they become more independent and reliable [63,64,65,66].

The automation of radiomics analysis could eventually allow for widely available clinical implementation of CNNs, minimizing the cost and duration of diagnosis for some tumor features, with experimental models already showing excellent performance [60]. Chang et al. trained a CNN to classify IDH1 mutation status, 1p/19q codeletion and MGMT promoter methylation status in gliomas, achieving high accuracies of 94%, 92% and 83%, respectively, in addition to specifying the key imaging features critical for effective classification [67]. Luckett et al. also recently trained a deep neural network to classify overall survival (OS) in glioblastoma patients by looking at demographics and multimodal neuroimaging features. The results were over 90% accurate, also proving that machine learning holds the potential to account for glioblastoma’s global effect on brain structural and functional organization, which is predictive of survival [68]. Deep learning models are flexible and can discriminate high-level features such as molecular characteristics directly from images, with the caveat that they require more samples than standard machine learning models [69].

The use of ADC values for determination of MGMT methylation status has historically yielded inconclusive results [70,71,72], and even the same patient cohorts can yield different metrics depending on the method of data analysis used [73]. However, since the incorporation of machine learning and deep learning networks to analyze MRI features, increasingly better results than through manual radiological analysis have been achieved [74,75]. Yogananda et al. recently used a deep learning network that only analyzed T2WI sequences to classify MGMT promoter methylation status with high accuracy (94.73%), sensitivity (96.31%) and specificity (91.66%) [76].

Still, more studies need to be conducted using multicenter data and external validation before making these networks available in clinical settings. In their systematic review, Jian et al. pooled data from forty-four original articles using machine learning models, showing a sensitivity of 0.88 and specificity of 0.86 in predicting IDH mutation status and even lower values for MGMT and 1p/19q codeletion [77]; these results are promising but indicate the need for further optimization. Choi et al. later found no prognostic value for OS and progression-free survival (PFS) in a radiomics analysis of MR in glioblastoma [78]. A possible reason why data from different studies are so varied and difficult to compare is the differences in parameters used in each study, especially when using advanced MRI techniques such as APT CEST, for which there is no consensus protocol with respect to the standard parameters [79].

Investigations involving the prediction of other molecular markers for glioblastoma such as EGFR amplification and TERT promoter mutations using deep learning models have also shown correlations of different imaging parameters with increasingly better performance [80,81,82,83], and some studies have focused on elucidating gender variations in imaging characteristics and how such variations translate to molecular analysis, the timing of the acquisition of driver mutations, metabolism requirements, the immune landscape, therapeutic response and OS [84,85]. As demonstrated by Barnett et al., females present with MGMT methylation more often than males, in addition to showing greater PFS and OS benefit by MGMT methylation, which is not seen in males [86]. Therefore, the training of models by including gender as an independent parameter for diagnosis is justified.

In summary, radiogenomics research has confirmed that imaging features relate to molecular features better than they relate to histologic characteristics. CNNs can also both learn and teach us how to look at images in the hopes of enabling more accurate diagnosis and decreasing the need for invasive procedures [61]. Nonetheless, larger multicenter studies are necessary to improve the performance of deep-learning models, which could undoubtedly facilitate the widespread adoption of advanced MRI techniques and protocols in clinical settings.

## 5. Molecular Features of Glioblastoma and Prognostic Implications

Molecular biomarkers provide measurable molecular indicators of the risk of cancer development, cancer recurrence and patient outcome. Such biomarkers can include genetic variants, epigenetic signatures, transcriptional changes and proteomic signatures [87] (Figure 1). The current WHO CNS5 classification puts great emphasis on molecular markers including IDH mutation status, EGFR amplification, TERT promotor mutation and +7/−10 chromosome copy-number variations, which are mainly used for diagnosis and classification (Figure 1).

Glioblastoma can carry around 60 mutations [56], with most of them being passengers and only a few being actual driver mutations that confer a selective growth advantage and subsequent cancer progression [88]. Due to elevated tumor heterogeneity, even within the same tumor, being able to distinguish driver mutations could help to identify therapeutic targets and a more adequate course of treatment, which has prompted development of machine learning-based methods for the identification of driver mutations [89].

Lee et al. [15] reported that astrocyte-like NSCs in the SVZ contain driver mutations of human glioblastoma. Utilizing a combination of patient brain tissue and genome editing of a mouse model, they demonstrated that in glioblastoma, IDH-wildtype driver mutations seen in the main tumor matched those seen in corresponding SVZ tissue. Furthermore, they showed that the NSCs that harbor driver mutations migrate from the SVZ to distant brain regions, where they clonally evolve and develop the tumor. Most mutations shared between the tumor and SVZ were found to be TERT promoter or cancer-driving genes, such as *EGFR*, *PTEN* and *TP53* [15].

Glioblastoma’s genetic heterogeneity makes it easy for somatic aberrations to flourish abundantly, but no single mutation has been proven to individually trigger tumorigenesis, which limits the clinical relevance of most of them. A multi-omics analysis conducted by Herrera-Oropeza et al. [90] suggested that a combination of these biomarkers can provide a multidimensional approach that leads to better diagnosis, as well as glioblastoma molecular subtype classification for prognosis. Their analysis also highlighted the complexities and challenges linking phenotypic alterations with the expressed tumor genotype, since epigenetic changes. as well as distant genetic interactions, can influence gene expression in the absence of mutations. In this section we discuss some of the most common molecular biomarkers used in glioblastoma and how do they translate to current clinical practices.

### 5.1. IDH Mutation Status

Since the publication of the WHO CNS5 in 2021, by definition, a glioblastoma is a glioma whose *IDH* gene is in its wildtype form [6]. The *IDH* gene provides the instructions to produce the IDH enzymes that convert isocitrate to α-ketoglutarate, yielding an NADPH molecule, which helps protect cells from ROS [56] (Figure 2). Notably, wildtype IDH1 is overexpressed in glioblastoma, an adaptation that supports macromolecular synthesis, aggressive growth and therapy resistance [91]. Since wildtype IDH is involved in many metabolic processes, it is thought to be a key driver of tumor oncogenesis. However, approaches for therapies targeting this gene require further investigation, especially since the main function of IDH enzymes, the oxidative decarboxylation of isocitrate to α-ketoglutarate, is displayed by both normal and tumoral cells, making it hard to target [92]. Selective inhibition of IDH in tumor cells could prevent tumor growth, reduce the frequency of glioblastoma stem-like cells involved in recurrence and synergize with chemoradiation. Inhibitors of mutant IDH1 have been characterized, such as the recently FDA-approved Vorasidenib [93], a first-in-class dual inhibitor of mutant IDH1 and IDH2 [94] with proven efficacy in IDH-mutant gliomas. However, the development of specific inhibitors for wildtype IDH1 is limited and mostly restricted to the lab [95]. Some therapies developed to inhibit mutant IDH may also inhibit wildtype IDH1 activity to some extent. Furthermore, tumor heterogeneity, the wide range of metabolic processes and locations of IDH enzymes and the high doses required for these therapies to be effective impact clinical utility in glioblastoma. Targeting IDH enzymes via small molecules instead of solely targeting genomic alterations has also been addressed. For instance, in a murine xenograft model of human glioblastoma, IDH1 silencing improved the response to fractionated radiotherapy via the reduction of NADPH, deoxynucleotides and glutathione, which usually help repair radiation-induced DNA damage [96] (Figure 2). Targeting the specific IDH3α subunit usually upregulated in glioblastoma through CRISPR/Cas9 silencing can disrupt nucleotide biosynthesis, rendering cells vulnerable to antifolate therapy, such as methotrexate, and subsequent programmed cell death [97] (Figure 2). These approaches should be further investigated, considering them in combination with other targeted therapies.

### 5.2. TERT Promoter Mutation

The activity of human telomerase reverse transcriptase (TERT) compensates for the loss of telomere length that occurs with every cell cycle, ensuring appropriate cell division. In some tumors, like glioblastomas, alterations of the gene encoding TERT, such as TERT promoter (TERTp) mutations or methylation at the TERTp, can cause the overexpression of TERT and confer a proliferative advantage to neoplastic cells [98] (Figure 3). Mutations of TERTp can be expressed in over 80% of gliomas, making it the most prevalent non-coding mutation and indicating this as the primary mechanism of telomerase activation [99]. It is even theorized that TERTp mutations may be the earliest genetic event in NSCs in the SVZ, facilitating adaptive resistance to replicative senescence, and subsequently increasing the probability of the acquisition of driver mutations in NSCs and development into glioblastoma [15]. The clinical relevance of TERTp mutations is mainly for diagnostic and not for prognostic purposes in glioblastoma, since evidence of TERTp as a prognostic marker is divisive [99,100], with some studies stating that neither TERTp mutations nor telomerase length correlate with survival [101]. However, in an exploratory analysis of the SPARE trial, TERTp mutation was associated with better outcomes in patients treated with tumor treating fields [102].

TERTp mutation has been explored as a potential therapeutic target, given its ubiquitous presentation in glioblastoma (Figure 3). To this end, different mechanisms have been investigated as promising approaches, but limitations such as the toxicity of agents, difficult drug delivery, the absence of in vivo models for evaluation, and molecular and structural tridimensional data only being recently elucidated have hindered their clinical feasibility in glioblastoma [100]. Among the approaches that have reached the clinical phase, an oligonucleotide called imetelstat (or GRN163L), proven to inhibit TERT function in some hematological malignancies, failed in solid brain tumors due to hematologic dose-limiting side effects, with a phase II clinical trial (NCT01836549, https://clinicaltrials.gov/study/NCT01836549, accessed on 27 October 2024) needing to be terminated due to two patients dying of intratumoral hemorrhage secondary to thrombocytopenia [103].

Telomerase-based therapeutic cancer vaccines (TCVs) have been under clinical investigation for their potential to synergize with checkpoint inhibition, leading to enhanced immune response [104]. Specifically, there have been two noteworthy TCV therapeutic trials for glioblastoma. In a phase I/II trial (NCT03491683, https://clinicaltrials.gov/study/NCT03491683, accessed on 27 October 2024) [105], a DNA vaccine containing INO-5401, a synthetic DNA plasmid encoding human TERT, Wilms Tumor gene-1 (WT-1) and prostate-specific membrane antigen (PSMA) plus INO-9012 a synthetic DNA plasmid encoding IL-12, in combination with cemiplimab, a PD-1 inhibitor, was evaluated in two glioblastoma cohorts (A: unmethylated MGMT; B: methylated MGMT). The trial results showed a safe risk/benefit profile with robust immune responses and enhanced survival when administered with protocol radiotherapy/temozolomide in newly diagnosed glioblastoma patients. The median OS in cohorts A and B was 17.9 months and 32.5 months, respectively [105]. In another phase IIa trial, UCPVax, a vaccine composed of two CD4w helper peptides derived from TERT (NCT04280848, https://www.clinicaltrials.gov/study/NCT04280848, accessed on 27 October 2024), was evaluated in 31 glioblastoma, IDH-wildtype, MGMT-unmethylated patients and proved to be highly immunogenic with a relatively safe profile, providing an improved OS rate in this population, which presents a rationale for further clinical studies of UCPVax in glioblastoma patients, possibly in combination with other therapies [106]. A comprehensive review on current vaccine clinical trials for glioblastoma was published by Xiong et al. and is recommended for further reading [107]. In Table 2, we list the results of these studies, along with further drugs currently being investigated.

Nucleoside analogues such as 6-thio-2′-deoxyguanosine (6-thio-dG) or 5-fluoro-2′-deoxyuridine (5-FdU) triphosphate can induce telomere dysfunction and cell death when incorporated into newly synthesized telomeres [115,116]. Mender et al. showed that cells treated with 6-thio-dG undergo telomere stress with subsequent DNA damage that is sensed by dendritic cells, activating innate and adaptative immune-dependent responses, which could open the path for the combination of telomere-targeted therapy with immunotherapies like anti-PD-L1 [108]. However, these effects have not been tested in glioblastoma cells.

BIBR1532, a telomerase inhibitor, showed a dose-dependent cytotoxic effect in cultured human glioblastoma cells after treatment, with reduced expression of TERT protein [117]. However, BIBR1532 is highly insoluble in water, limiting its therapeutic availability due to its low cellular uptake and inadequate delivery. To overcome this issue, zeolitic imidazole framework-8 (ZIF-8) was tested as a drug delivery vehicle, improving the transportation and release of BIBR1532 to the nucleus, inhibition of TERT mRNA expression, cell cycle arrest and cellular senescence in treated cancer cells when compared with free BIBR1532 [109]. This combination is yet to be tested in human glioblastoma cells.

G-quadruplexes (G4), guanine-rich sequences forming secondary structures frequently occurring in telomeres, can interfere with normal TERT function, causing DNA damage, cell cycle arrest, apoptosis and senescence. Therefore, specific ligands that stabilize G4 are potentially cytotoxic. Telomestatin is a naturally occurring compound that stabilizes G4 and preferentially impairs the growth of glioma stem-like cells (GSCs) [118]. However, the mechanism for the GSC-selective nature of the DNA damage response remains unknown [118]. A synthetic G4 ligand, a Y2H2-6M(4)-oxazole telomestatin derivative (6OTD), was tested in human glioblastoma cancer cells and mouse xenografts, demonstrating the inhibition of growth of GSCs and DNA damage, preferentially in telomeres of GSCs, which contain repetitive G4-forming sequences [111]. BRACO-19, a specific ligand for telomeric G4 showed decreased viability of human glioblastoma cells while sparing normal surrounding cells in a highly specific manner [110], which could serve as the rationale for using this G4 ligand to sensitize glioblastoma cells to other therapies. Berardinelli et al. combined another G4 ligand, RHPS4, with ionizing radiation therapy in a heterotopic mouse xenograft model of differentiated glioblastoma cells and in vitro in stem-like cells derived from glioblastoma patients [112]. Although radiosensitization was achieved in the differentiated cells, no synergistic radiosensitization was achieved in GSCs despite the inhibition of GSC growth, suggesting that the RHPS4 effects on GSC growth inhibition are not mediated by telomeric dysfunction as in their differentiated counterparts but through the induction of replication stress with the depletion of S-phase proteins and subsequent DNA damage [112].

Genetic therapies using mechanisms such as CRISPR and programmable base editing can be used to silence TERTp or associated genes required for telomere maintenance. For example, local injection of adeno-associated viruses expressing an sgRNA-guided and catalytically impaired *Campylobacter jejuni* CRISPR-associated protein 9-fused adenine base editor (CjABE) reverts TERTp mutation, reducing TERT transcription and TERT protein expression and subsequently inhibiting glioma growth [113]. Another example is a shRNA specifically targeting the GABPβ1L subunit of the GA-binding protein (GABP) transcription factor, which binds to mutated TERT promoters and allows for continuous TERT reactivation and telomere maintenance. Targeting GABPβ1L decreases TERT expression, leading to synergistic inhibition of tumor growth when combined with temozolomide [114].

The fact that TERT inhibition is still far from being clinically feasible and the latency time required to achieve biological effects are limitations to consider. However, novel approaches using combination strategies, such as immunotherapy agents, in particular, are worth exploring further.

### 5.3. EGFR Gene Amplification

The epidermal growth factor receptor (EGFR), also referred to as HER1/ErbB1, is a transmembrane receptor with tyrosine kinase activity with other members of the HER family, including ErbB2/HER2, ErbB3/HER3 and ErbB4/HER4. EGFR is involved in two key downstream signaling pathways: the PI3K/AKT/mTOR pathway and the Ras/Raf/MAPK pathway, which promote proliferation, cell survival and migration [5,119]. EGFR was one of the first biomarkers studied in glioblastoma, and the presence of amplification is almost exclusively seen in glioblastoma, making its detection influential for diagnosis [3,120]. A meta-analysis published by Li et al. evaluated 17 articles containing 1458 patients and found that the overexpression of EGFR is an indicator of poor prognosis for glioblastoma patients [121]. The EGFR protein is frequently overexpressed in up to 90% of glioblastomas, with the most common anomaly being an increased number of copies of the gene by focal gene amplification, which is observed in around 40% of cases. EGFR amplification occurs mostly in the infiltrating edges of tumors and is considered to play a role in gliomagenesis and tumor invasiveness [56]. Mutant variant EGFRvIII represents a truncated yet constitutively active form of the receptor that allows for cell proliferation and tumor resistance to therapy [122,123,124]. EGFRvIII is highly expressed in glioblastoma as a cancer-specific driver mutation and, hence, is considered an ideal therapeutic target [122,123,124]. Targeted therapies against EGFR have showed efficacy in other cancers, but their use in glioblastoma has yielded varied results, with some of their limitations including difficult blood–brain barrier (BBB) penetration and intrinsic resistance [124]. The results of selected studies for these therapies are summarized in Table 3.

Small-molecule tyrosine kinase inhibitors (TKI) such as erlotinib, gefitinib and lapatinib preferentially target mutations in the extracellular domain of the EGFR tyrosine kinase receptor when it is conformationally active. However, in glioblastoma, EGFR receptor mutations are primarily intracellular and display active signaling but are conformationally inactive; hence, these traditional TKIs are not completely effective in targeting EGFR in glioblastoma [124]. Strategies have been proposed to tackle this issue, one being the combination of TKIs with other therapies in the hopes of making them work synergistically, but results in humans have failed to meet the primary endpoint of increased survival. For example, a phase II study (NCT00720356, https://clinicaltrials.gov/study/NCT00720356, accessed on 27 October 2024) combining erlotinib with bevacizumab after the Stupp protocol in MGMT-unmethylated glioblastoma patients showed no improvement in OS [125]. Conversely, a small cohort study [126] evaluating the efficacy of TKI osimertinib plus bevacizumab and standard therapy in glioblastoma patients with EGFR amplification and EGFRvIII mutation showed marginal effectiveness in PFS and OS; however, a small subset of patients actually developed meaningful benefit, which could prompt further studies on the specific characteristics that made them more responsive to therapy.

Improving the BBB penetration of these drugs without causing systemic toxicity is another limitation that is under investigation. Oguchi et al. trialed TAS2940, a novel, irreversible pan-ErbB inhibitor with remarkable brain penetration, in generic cell lines and intracranial mouse xenograft cancer models, demonstrating inhibition of tumor growth against cells with HER2 amplification and EGFRvIII mutation and improving the survival of subjects, indicating its therapeutic potential for glioblastoma with EGFR mutations [127]. A phase I trial testing TAS2940 in solid tumors with EGFR and/or HER2 aberrations is ongoing (NCT04982926, https://clinicaltrials.gov/study/NCT04982926, accessed on 27 October 2024) and currently recruiting.

The design of new, highly specific antibodies targeting EGFR in glioblastoma has also been addressed thoroughly. One of the first monoclonal antibodies (mAbs) designed with this purpose is ABT-806, which binds to an exposed epitope in the EGFRvIII mutant receptor, demonstrating inhibition of growth of xenograft models [141]. Its derivative, depatuxizumab mafodotin (Depatux-m) was tested in a phase III clinical trial versus placebo, showing increased PFS but no improvement in OS in newly diagnosed glioblastoma patients, with EGFR and corneal epitheliopathy occurring as an adverse effect in 94% of Depatux-m-treated patients [128].

Intravenous mAb GC1118 was evaluated in a multicenter, open-label, single-arm phase II trial (NCT03618667, https://clinicaltrials.gov/study/NCT03618667, accessed on 27 October 2024) in recurrent glioblastoma patients with EGFR amplification; however, despite being relatively well tolerated, it did not show benefit in PFS or OS [129]. Panitumumab is another mAb with clinical potential yet to be tested further, having been shown to exhibit superior antitumor activity in vitro and in vivo compared to other mAbs due to its ability to neutralize both EGFRvIII and wildtype EGFR activation [142]. In the ongoing DRUP trial (NCT02925234, https://clinicaltrials.gov/study/NCT02925234, accessed on 27 October 2024), a prospective, multicenter, non-randomized basket and umbrella trial, patients are enrolled in multiple parallel protocols depending on molecular alterations and study drugs [143]. Panitumumab was investigated in a cohort with RAF/RASwt glioblastoma that received this drug intravenously every 2 weeks until disease progression or intolerable side effects occurred. While 24 patients were treated and the drug was found to be safe to use, the overall clinical benefit was limited, with only three patients showing stable disease after 16 weeks [130]. A phase I/II clinical trial (NCT03510208, https://clinicaltrials.gov/study/NCT03510208, accessed on 27 October 2024) repurposing panitumumab as an optical imaging agent when combined with optical dye IRDye800CW in malignant gliomas is ongoing.

Nimotuzumab, an mAb that targets the extracellular domain of EGFR, was tested in a phase II, single-arm, multicenter clinical trial (NCT03388372, https://clinicaltrials.gov/study/NCT03388372, accessed on 27 October 2024) to evaluate the benefit of adding it to standard therapy, and the results showed increased survival in newly diagnosed glioblastoma patients with positive EGFR expression [131]. More recently, the combination of nimotuzumab with melatonin, which is known to disrupt EGFR in its intracellular segment, demonstrated increased cytotoxicity and apoptosis of cancer cells in vitro and in xenograft mouse glioblastoma models [132]. Although these results still need clinical translation in humans, this approach stresses circadian rhythms and sleep disorders as factors to be considered in cancer therapies.

Rindopepimut, a peptide-based vaccine, targets EGFRvIII mutation exclusively in glioblastoma. Rindopepimut was evaluated in a randomized, double-blinded, international phase III clinical trial, ACT IV, in which 745 patients with glioblastoma were enrolled to receive either rindopepimut and temozolomide or temozolomide alone; however, the study was terminated after showing futility in an interim analysis, with no significant difference in OS between groups [144]. When combined with bevacizumab in a double-blind, randomized phase II study (NCT01498328, https://clinicaltrials.gov/study/NCT01498328, accessed on 27 October 2024) in recurrent EGFRvIII-expressing glioblastoma patients, the rindopepimut arm showed increased PFS at six months and a survival advantage over the control group; however, validation of these findings with bigger sample sizes is required [133].

Oncolytic virus (OV) therapy consists of infecting tumor cells with a virus that triggers immunogenic tumor cell death [145]. OVs that have demonstrated efficacy against glioblastoma in preclinical studies include adenovirus, herpes simplex virus, measles virus, parvovirus, poliovirus and zika virus [146]. Most studies have evaluated the efficacy of OVs alone and in combination with other immunotherapies. On its own, R-613, an oncolytic herpes simplex virus (oHSV) retargeted to EGFRvIII, delayed the development of tumor masses and increased OS in orthotopically transplanted mice when given as an early treatment [134]. An oHSV engineered to express EGFR antibody cetuximab linked to the chemotactic chemokine CCL5 enhanced chemotactic and activation immune cells, inhibited tumor EGFR signaling, reduced tumor size and prolonged survival in glioblastoma-bearing mouse models [135]. Similarly, combining an oHSV-expressing IL-15/IL-15Rα with CAR NK cells resulted in the synergistic inhibition of tumor growth and increased survival in mice [136]. Promising results were also seen in a fully virulent oHSV retargeted to human ErbB-2 to deliver murine IL-12 in murine glioblastoma models, with an unprecedented complete eradication of the tumor in 30% of subjects, which is something rarely seen with the most commonly used mutated or attenuated OVs [137].

EGFRvIII is a compelling target for CAR T cells, and several studies have examined the efficacy of these cells both alone and in combination with other therapeutics. A phase I trial (NCT01454596, https://clinicaltrials.gov/study/NCT01454596, accessed on 27 October 2024) assessed the safety and PFS of anti-EGFRvIII CAR T cells in 18 glioblastoma patients, but no clinically meaningful effect was seen [138]. Another phase I trial (NCT03726515, https://clinicaltrials.gov/study/NCT03726515, accessed on 27 October 2024) evaluated concomitant administration of anti-EGFRvIII CAR T cells and pembrolizumab in seven patients with newly diagnosed EGFRvIII mutant glioblastoma; however, despite the upregulation of the tumor microenvironment, no improvement was demonstrated in terms of PFS or OS [139]. IL13Rα2 is an innovative target for CAR T cells. Intrathecally delivered CAR T cells targeting EGFR and IL13Rα2 were evaluated in a phase I trial (NCT05168423, https://clinicaltrials.gov/study/NCT05168423, accessed on 27 October 2024) in six patients with glioblastoma recurrence, demonstrating safety with only early-onset neurotoxicity consistent with immune effector cell-associated neurotoxicity syndrome (ICANS), which was clinically manageable, and moderate efficacy, with reductions in enhancement and tumor size detected by MRI in all six patients [140].

The relevance of EGFR mutations in glioblastoma is strong, but despite therapeutic potential, no single targeted therapy against EGFR or its variants has been proven to improve survival in glioblastoma patients. However, the use of combinations that can target different tumor markers at the same time should be further researched.

### 5.4. Concomitant Chromosome 10 Loss and Chromosome 7 Gain

The combined losses of chromosome 10 and gains of chromosome 7 gain are a characteristic molecular alteration in IDH-wildtype glioblastoma that is most likely a result of errors in mitotic disjunction [5,6,147]. In some study cohorts, as well as the TCGA cohort, the presence of at least one clonal copy number of either of these alterations was found in all glioblastoma tumors, suggesting that they are driver mutations for early tumorigenesis [3,148].

The loss of chromosome 10 can be seen in up to 80% of glioblastoma cases [149] and is commonly linked to the inactivation of phosphatase and tensin homolog (*PTEN*), a tumor suppressor gene located on 10q23 that inhibits the transduction of growth factor signals through the PI3K/AKT/mTOR pathway; regulates cell migration; and, more importantly, triggers apoptosis, blocking tumorigenesis [56]. The loss or downregulation of *PTEN* has been associated with poor prognosis in glioma patients [150], especially in patients over 45 years old with homozygous *PTEN* deletion [151]. However, in a meta-analysis of over 14,678 patients, loss of heterozygosity of chromosome 10/10q did not appear to have any significant prognostic value in glioblastoma, despite its role in gliomagenesis, rendering its role as a biomarker controversial in clinical practice [152].

Gains in chromosome 7 are also common in glioblastomas and are associated with mutations or amplification of oncogenes such as *EGFR* and *MET* [5,153]. These alterations are usually associated with monosomy of chromosome 10, probably due to the upregulation of rescuer genes on chromosome 7 [149,154]. *EGFR*-targeted therapies were discussed in a previous section, but we briefly discuss recent investigations of *MET*-targeted therapies below. The *MET* gene transcribes the Met tyrosine kinase receptor for the hepatocyte growth factor/scatter factor (HGF/SF), which promotes signaling cascades that, under normal conditions, modulate epithelial-to-mesenchymal transition in tissue repair and embryogenesis. However, in cancer, Met activity promotes the proliferation, survival and migration of tumors, and in glioblastoma, MET overexpression is linked to resistance to chemotherapy [153].

Several small-molecule inhibitors have emerged to target oncogenic *MET* aberrations in glioblastoma. Cabozantinib is a TKI with activity against VEGFR2 and MET and demonstrated effective activity against various solid tumors [155]. In a phase II trial (NCT00704288, https://www.clinicaltrials.gov/study/NCT00704288, accessed on 27 October 2024) conducted to evaluate its effectiveness in recurrent glioblastoma, among the 152 patients enrolled, PFS at 6 months was 27.8% and OS was 10.4 months in patients receiving an adjusted decreased dose due to initial toxicity; however, the statistical target for success was not met [156]. A phase I/II trial assessing the combination of cabozantinib with atezolizumab, a mAb-targeting PD-L1, is currently ongoing (NCT05039281, https://clinicaltrials.gov/study/NCT05039281, accessed on 27 October 2024). Capmatinib (INC280), an oral ATP-competitive and highly potent MET inhibitor, was tested in a phase I clinical trial (NCT01324479, https://clinicaltrials.gov/study/NCT01324479, accessed on 27 October 2024), and the results showed an acceptable safety profile and antitumor activity in certain MET-positive solid tumors, including glioblastoma [157]. Its combination with bevacizumab is currently the subject of a phase I trial (NCT02386826, https://clinicaltrials.gov/study/NCT02386826, accessed on 27 October 2024) in patients with recurrent and unresectable glioblastoma. Crizotinib, an ALK, *ROS1* and c-MET inhibitor, was added to standard chemoradiation in a phase Ib, open-label, single-arm, multicenter study (NCT02270034, https://clinicaltrials.gov/study/NCT02270034, accessed on 27 October 2024) that enrolled a total of 38 patients; the results demonstrated a median PFS of 10.7 months,an OS of 22.6 months, and relative safety for patients [158]. These encouraging results should be validated in a powered, randomized, controlled study [158]. MET/HGF-specific mAbs such as onartuzumab have been investigated, with no clear clinical benefit shown in glioblastoma patients [159]. Table 4 summarizes selected therapies against the MET signaling pathway.

Some authors have stated that PTEN can determine the response to certain therapies. For example, Ma et al. identified a mechanism of resistance to ionizing radiation therapy in glioblastomas with FGFR2-mediated phosphorylation of PTEN on tyrosine 240 (pY240-PTEN) and found that blocking Y240 phosphorylation increased sensitivity to radiation and extended survival in mouse models, which could be a therapeutic approach to study further [160]. The results of attempts to target PTEN or the signaling cascades it regulate with various approaches are summarized in Table 5.

The use of gene therapies to modify *PTEN* expression is a fairly recent field of investigation, and the use of OVs to help upregulate its expression is being tested. *PTEN* adenovirus was used in the lab to overexpress *PTEN* in glioblastoma cells in vitro, showing increased cell apoptosis by mediating mitochondrial dysfunction and subsequent activation of mitochondrial apoptosis through a mechanism dependent on the activation of Drp1-related mitochondrial division via Akt pathway modulation [171]. Nan et al. showed that the combination of an adenovirus-mediated *PTEN* plus PI3K inhibitor LY294002 suppressed cell proliferation, arrested the cell cycle, reduced cell invasion and synergistically promoted cell apoptosis in in vitro and in vivo xenograft glioblastoma mouse models by effectively inhibiting the PI3K/AKT pathway, a promising result that warrants further investigation gene therapy [161]. Nevertheless, most tested therapies, both in preclinical and clinical models, have preferentially targeted the PI3K/Akt/mTOR signaling cascade.

Ipatasertib is a novel, potent, selective small-molecule inhibitor of Akt that has been tested in combination with atezolizumab, the mAb against PD-L1 in solid tumors, including glioblastoma, with the aims of depleting the tumor microenvironment of suppressive immune cells with Akt inhibition and of making tumor cells more responsive to immune checkpoint inhibitors. In a phase I/II open-label, single-center trial (NCT03673787, https://clinicaltrials.gov/study/NCT03673787, accessed on 27 October 2024), patients with recurrent glioblastoma were enrolled into two cohorts to assess the combination for safety and tolerability (cohort A2), as well as preliminary efficacy (cohort B3). Results showed the combination to be safe and well tolerated, with a clinical benefit (either complete response, partial response or stable disease at 6 months) in 32% in all patients and in 28.6% of patients with *PTEN* loss, which is theorized to be a promising predictive biomarker to assess the response to the combination [162,163].

PI3K signaling is highly active in glioblastoma, and multiple therapies targeting this specific part of the pathway are being investigated. Buparlisib, a PI3K inhibitor was tested in a phase II trial (NCT01339052, https://clinicaltrials.gov/study/NCT01339052, accessed on 27 October 2024) in patients with recurrent glioblastoma; however, it showed minimal efficacy despite significant brain penetration [164]. A multicenter phase Ib/II trial (NCT01934361, https://clinicaltrials.gov/study/NCT01934361, accessed on 27 October 2024) tested the combination of buparlisib plus either carboplatin or lomustine. The study did not demonstrate sufficient antitumor activity compared with data on single-agent carboplatin or lomustine [165]. Similarly, a phase Ib/II study (NCT01870726, https://clinicaltrials.gov/study/NCT01870726, accessed on 27 October 2024) comparing the efficacy of INC280 (capmatinib), a MET inhibitor, alone with its combination with buparlisib in adult patients with recurrent glioblastoma showed no clear activity using the combination, despite the described synergy in preclinical glioblastoma models [166]. Other clinical trials (NCT01349660, https://clinicaltrials.gov/study/NCT01349660, accessed on 27 October 2024, NCT01473901, https://clinicaltrials.gov/study/NCT01473901, accessed on 27 October 2024) combining buparlisib with bevacizumab [167] and buparlisib with standard chemoradiation [168] have rendered similar limited antitumor efficacy and intolerable side effects.

In patient-derived glioblastoma cells and orthotopic glioblastoma mouse models, Noch et al. [169] found that reducing insulin feedback with metformin and a ketogenic diet improves the treatment efficacy of PI3K inhibitors, given that PI3K inhibition induces hyperglycemia and hyperinsulinemia in mice. They also retrospectively examined blood and tumor tissue from patients in a phase II buparlisib trial and found that hyperglycemia is an independent factor that worsens PFS in these patients, which could be interpreted as hyperglycemia being a resistance mechanism for PI3K inhibition [169]. A phase II clinical trial is currently being conducted (NCT05183204, https://clinicaltrials.gov/study/NCT05183204, accessed on 27 October 2024), combining the PI3K/mTOR inhibitor paxalisib with a ketogenic diet plus metformin in patients with newly diagnosed and recurrent glioblastoma.

The combination of EGFR inhibitor AZD-9291, which blocks the EGFR/MEK/ERK pathway, with PI3K inhibitor GDC-0084, which blocks the PI3K/AKT/mTOR pathway, also recently showed synergistic inhibition of proliferation and survival in in vitro and in vivo models; therefore, it should be considered for future clinical trials [170].

Overall, the concomitant loss of chromosome 10 and the gain of chromosome 7 plays an important role in tumorigenesis due to the various critical genes on these chromosomes. The loss of tumor suppressor genes such as *PTEN* in chromosome 10, in particular, represent a well described mechanism of tumor evolution, and in recent years, many targeted therapies, such as the use of repurposed viruses, small-molecules inhibitors and monoclonal antibodies, have been focused on either increasing *PTEN* expression or blocking the pathways it naturally downregulates, with mostly discouraging outcomes. However, investigation is still ongoing, and therapies against other molecules such as MET have shown some promising results in recent clinical trials.

### 5.5. MGMT Promoter Methylation Status

Methylation of the *MGMT* gene promoter occurs in 35–45% of malignant gliomas [56], and it has been associated with decreased expression of O6-methylguanine-DNA methyltransferase (MGMT), an enzyme that reduces DNA damage [172]. Therefore, the methylation of the MGMT promoter enhances sensitivity to alkylating agents such as temozolomide [173] and prolonged OS and PFS in comparison to unmethylated MGMT glioblastoma [174,175], a rationale for its use as a biomarker in glioblastoma.

Therapeutic strategies in glioblastoma have focused on blocking MGMT activation, either through epigenetic agents that modulate the methylated state of the *MGMT* gene promoter or through direct inhibition of the MGMT protein (Figure 2). Table 6 summarizes some studies that have tested different therapies against MGMT activation.

CRISPR-based targeted methylation of the MGMT promoter, using a chimeric fusion protein “d3A” consisting of a deactivated Cas9 (dCas9) with an epigenetic editor (DNA methyltransferase 3A catalytic domain), has been proven to downregulate MGMT expression and enhance susceptibility to temozolomide in several studies using glioblastoma cell lines [176,177]. Making modifications at the epigenetic level only has the advantages of being specific and reversible and yields minimal off-target effects [176,177]. However, this methylation editing technology still needs some refinement before entering the clinical stage, and the use of nanocapsules for effective delivery to glioblastoma tissue is under investigation [184].

A phase I clinical trial (NCT01700569, https://clinicaltrials.gov/study/NCT01700569, accessed on 27 October 2024) carried out in France evaluated the safety and efficacy of administering folic acid combined with temozolomide and radiotherapy in unmethylated MGMT promoter glioblastoma patients, with folate acting as a methyl donor and increasing DNA methylation. The trial showed that the combination restored methylation of the promoter in samples of circulating tumor DNA of eight patients, with a well-tolerated safety profile [178]. These results are in concordance with a phase I/II trial (NCT01891747, https://clinicaltrials.gov/study/NCT01891747, accessed on 27 October 2024) whose phase I results showed a 25% increase in DNA methylated CpGs in tumor autopsies compared with the paired initial recurrent glioblastoma IDH-wildtype tumors [179]; nevertheless, the clinical efficacy of adding folate is yet to be determined.

Bypassing *MGMT* gene expression and directly targeting the activated MGMT protein has also been explored. Lomeguatrib, a highly specific MGMT inhibitor, has been proven to inactivate the MGMT protein in vitro [185]. Kirstein et al. demonstrated that the inhibition of MGMT by lomeguatrib enhances radiosensitivity in MGMT-producing glioblastoma cell lines, as well as MGMT silencing using siRNA [180]. A recent study proved that epigenetic reactivation of Tumor Suppressor Candidate 3 (TUSC3) can reprogram the sensitivity of GSCs to temozolomide, irrespective of MGMT promoter methylation status [181]. Moreover, the study provided evidence that demethylating agent 5-Azacitidine (5-Aza) can reactivate TUSC3 expression in MGMT-methylated GSCs, whereas 5-Aza needs to be combined with MGMT inhibitor lomeguatrib in MGMT-unmethylated GSCs, as demonstrated in orthotopic GSCs models [181]. Lomeguatrib is a drug that holds potential for future clinical studies.

Pre-treatment with 26S-proteasome inhibitor bortezomib prior to temozolomide caused MGMT mRNA and protein depletion in glioblastoma cell lines, causing cell death in vitro and prolonging survival in orthotopic mice [182]. A phase Ib/II trial (NCT03643549, https://clinicaltrials.gov/study/NCT03643549, accessed on 27 October 2024) is investigating the safety and survival benefits for patients with recurrent grade-4 gliomas with unmethylated MGMT promoter treated with bortezomib and temozolomide [186]. Marizomib, a novel proteasome inhibitor, crosses the blood–brain barrier, which is an advantage over bortezomib [187]. Marizomib has been tested in multiple clinical trials (NCT02330562, https://clinicaltrials.gov/study/NCT02330562, accessed on 27 October 2024, NCT02903069, https://clinicaltrials.gov/study/NCT02903069, accessed on 27 October 2024, and NCT03463265, https://clinicaltrials.gov/study/NCT03463265, accessed on 27 October 2024) in glioblastoma patients in combination with standard and newer therapies [188,189,190]. However, a phase III multicenter, randomized, controlled, open-label trial (NCT03345095, https://clinicaltrials.gov/study/NCT03345095, accessed on 27 October 2024) showed more toxicity and no statistically significant difference in OS or PFS between patients receiving marizomib in addition to standard treatment (RT/Temozolomide) and patients receiving standard treatment alone [183].

Although clinically relevant for prognosis and deciding on patient management, the methylation of the MGMT promoter and the MGMT protein itself are yet to be considered feasible targets for therapy. Some branches of investigation hold potential but require further research to be proven beneficial.

### 5.6. Other Potential Targets for Therapy

As previously stated, glioblastoma is a very heterogeneous tumor type. A wide array of genes can be expressed in different individual tumors [90], and their use as targets for therapy has been explored to a certain extent (Figure 1).

BRAF V600E is a mutation with a 1–3% prevalence in glioblastoma [191]. It causes constitutive activation of the downstream effector cascade of MAPKs, MEK1/2 and ERK1/2, which are regulators of cell survival and tumorigenesis. Combinations of BRAF/MEK inhibitors such as dabrafenib/trametinib showed promising clinical results in a case series [192].

Gene fusions of the neurotrophic receptor tyrosine kinase (*NTRK*) that code different tropomyosin receptor kinase (Trk) members of the receptor tyrosine kinase (TRK) family act as oncogenic drivers for several tumors, especially in children and rarely (<2%) including high-grade gliomas [193,194,195,196]. TRK regulates the RAS/MAPKs, PI3K/AKT and PLCγ pathways known to activate cell growth and proliferation, and *NTRK* fusions constitutively activate them [5,193]. TRK inhibitors are potential target therapy strategies, but clinical significance in glioblastoma is not yet clear, since most studies have not included CNS tumors [194]. A case report demonstrated that larotrectinib, a first-generation TRK inhibitor, was used to successfully treat an adult patient with a diffuse *NTRK3* fusion-driven high-grade glioma [195]. In contrast, a recent study showed failure of treatment with Entrectinib, another TRK inhibitor, in an adult patient with glioblastoma [197]. Previously, entrectinib demonstrated efficacy in an adult *NTRK2* fusion glioblastoma [198]. More information is needed to draw conclusions regarding the effectiveness of these therapies in glioblastoma patients and possible resistance mechanisms for treatment failure. Patwell et al. characterized a truncated splice variant of the TrkB.T1 receptor in gliomas [199]. They demonstrated that this variant is capable of activating the same cascading pathways as TrkB, the constitutively active kinase in *NRTK2* gene fusions, proposing expansion to the whole gene and gene fusion analysis to investigate possible mechanisms of resistance to TRK inhibitors [199]. Two clinical trials testing repotrectinib (NCT03093116, https://clinicaltrials.gov/study/NCT03093116, accessed on 27 October 2024, NCT04094610, https://clinicaltrials.gov/study/NCT04094610, accessed on 27 October 2024) in patients with solid tumors harboring an *ROS1*, *NTRK1*, *NTRK2* or *NTRK3* gene arrangements are currently recruiting.

## 6. Discussion and Future Perspectives

Advances in the molecular diagnosis of glioblastoma have been exponential in the past few years, to the point where it is now unthinkable to make a diagnosis without molecular data. With better understanding of molecular biomarkers and the intracellular pathways they control, the development of new technologies and therapies to target them will follow suit.

The use of AI and machine learning technology has helped the field of radiogenomics advance exponentially in the last five years, adding a new layer to our knowledge about molecular markers with machines. Scientists are able to comprehend more about the complex biology of glioblastoma, and machines can provide analysis of big data to discover new associations and possible avenues for treatment, with increasing accuracy and effectiveness. Multi-omics analysis is becoming essential for the development of targeted therapies, and as machines keep learning, more investigation in all areas is to be expected, including with respect to the diagnosis and treatment of glioblastoma.

We are entering the era of personalized treatments, and molecular biomarkers are the key to achieving this goal. Drugs and technologies specifically tailored to target a patient’s tumor could be the key to improving survival, and the most common molecular alterations of tumors have already been characterized and targeted. *EGFR* gene amplification is perhaps the most widely studied target, but most clinical trials have failed to improve survival rates. Elucidating reasons for these failures and what the resistance mechanisms of these new therapies are has been mostly speculative. Tumor heterogeneity makes it possible for resistance to arise from more than one specific tumor cell pathway. This is a gap in the literature that we suggest should be addressed in future investigations and that could potentially benefit from integration with multi-omics analysis.

Immunotherapies designed to target EGFR, such as monoclonal antibody nimotuzumab and CAR T-cell therapy, have shown promising results. We expect the trend of investigations focused on mechanisms of regulation of a patient’s immune system and modification of the tumor microenvironment to continue and evolve toward personalized treatment models and combined therapies. For example, the use of a vaccine carrying two plasmids encoding TERT and IL-12, in combination with a PD-1 inhibitor, showed enhanced survival of glioblastoma patients in a phase I/II clinical trial, as discussed previously in this review. This substantiates the value of continued investigation of different combination therapies.

Therapies involving OVs represent a rapidly evolving field of research, as they hold great potential for glioblastoma [134]. OVs that have reached clinical trial phases mainly focus on modifying the immune landscape of tumors and not directly attacking intrinsic tumor cell machinery. Recently, CAN-3110 (rQNestin), an oHSV targeting nestin in glioblastomas, was granted a fast-track designation by the FDA after a successful phase Ib trial [200,201]. Many other modified viruses are being tested in preclinical studies, targeting several different molecular markers specific to glioblastoma. This is an extensive field that could continue to be explored in future years, as viruses can help overcome several resistance mechanisms of glioblastoma tumor cells.

The use of nanoparticles as delivery vehicles to bypass the BBB in glioblastoma is a relatively new field. Farooq et al. conducted a systematic review of 10 clinical trials evaluating 225 glioblastoma patients treated with nanomedicine-based therapies [202]. The authors found an inferiority in terms of survival of these approaches when compared with bevacizumab. However, among the biggest limitations of the review were the limited number of studies and the heterogeneity in methodologies and nanoparticles used therein [202]. We point to this as a scope of investigation that needs to be further explored in future years, as advances in nanomedicine are promising, although gaps in the literature are still considerable. Understanding the molecular landscape of individual patients can potentially help improve nanoparticle delivery of drugs and the best combinations to reach brain tumor cells.

The small-molecule field was initially promising, but results have not been as encouraging. We can argue that more investigation of the specific mechanisms of resistance to these therapies could improve outcomes of molecules that still hold promise, such as ipatasertib against the PI3K/AKT/mTOR pathway or crizotinib, a c-MET inhibitor.

Somewhat more holistic approaches are also being investigated, and things like sleep and circadian rhythms are changing the paradigm for already established therapies like temozolomide chemotherapy, as the effectiveness and resistance of drugs can be affected by them. For newer experimental therapies such as PI3K inhibitors, control over metabolic factors like hyperglycemia have shown improved results, with these results expected to these translate to clinical trials.

## 7. Conclusions

Despite efforts to improve survival in glioblastoma, results are still dismal. The increased knowledge we now have about the molecular background of tumors has opened the door for targeted therapies that have the potential to more effectively treat patients, especially for tumors as heterogenous as glioblastoma. However, efforts are far from being translated to clinical practice, and many of the attempts made so far have not proven effective for a variety of reasons, including resistance mechanisms. Immunotherapies, which are currently at the forefront of novel therapeutic approaches for glioblastoma, offer hope in the field of targeted therapies. The development of combined therapies that can target several molecular pathways simultaneously while being effectively delivered to tumor cells without affecting normal brain tissue or causing dangerous side effects for the patients is what most research is trying to achieve. Other factors, such as the metabolic profile and circadian rhythms of tumors, have also recently been brought into the spotlight. With this review, we aimed to provide a comprehensive overview of the current literature on molecular biomarkers and precision therapeutic strategies in glioblastoma to condense the available information in one place and emphasize what is known and what needs to be further developed. Multi-omics analysis, resistance mechanisms and combination of therapies are three fields that can benefit from additional investigation. Integrating them could potentially help achieve the goal of providing a personalized treatment model for each patient and offer alternative approaches for tumors as complex as glioblastoma.

## Figures and Tables

**Figure 1 cancers-16-03635-f001:**
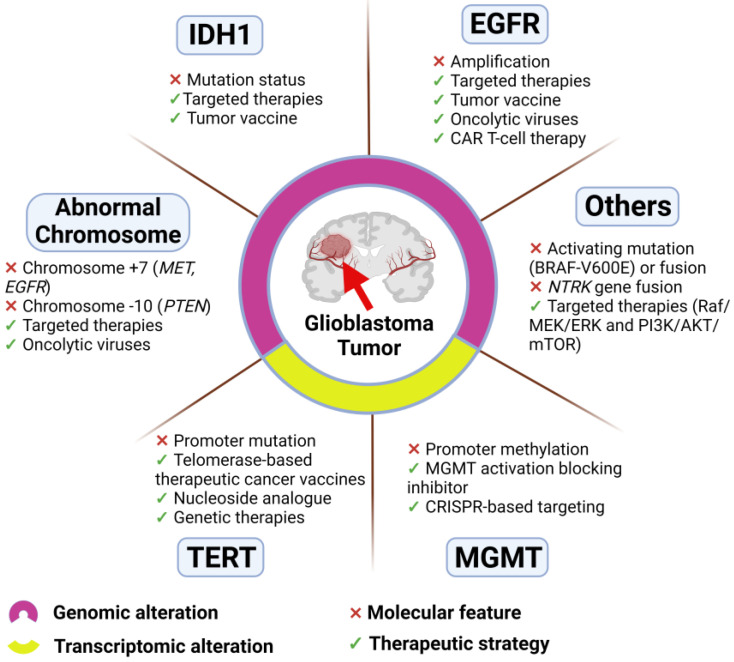
Molecular features of glioblastoma and related therapeutic strategies. Distinction indicates genomic (IDH1, EGFR, abnormal chromosome, and others) and transcriptomic (TERT and MGMT) alterations.

**Figure 2 cancers-16-03635-f002:**
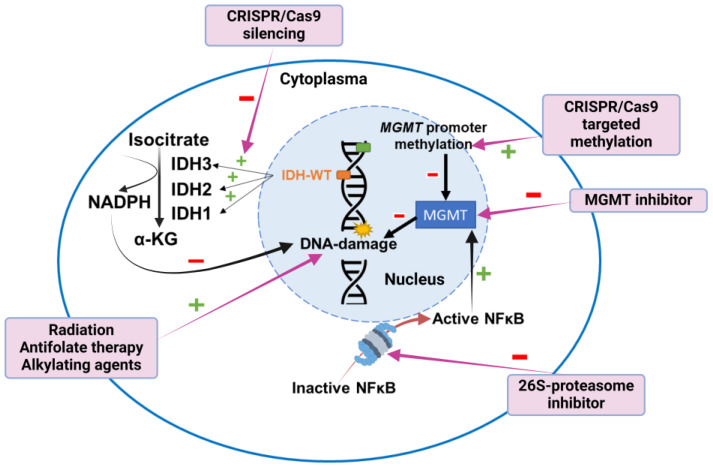
Schematic representation of mechanisms for DNA damage control via MGMT promoter methylation and IDH-wildtype status in glioblastoma. Different therapies affect these mechanisms at various sites.

**Figure 3 cancers-16-03635-f003:**
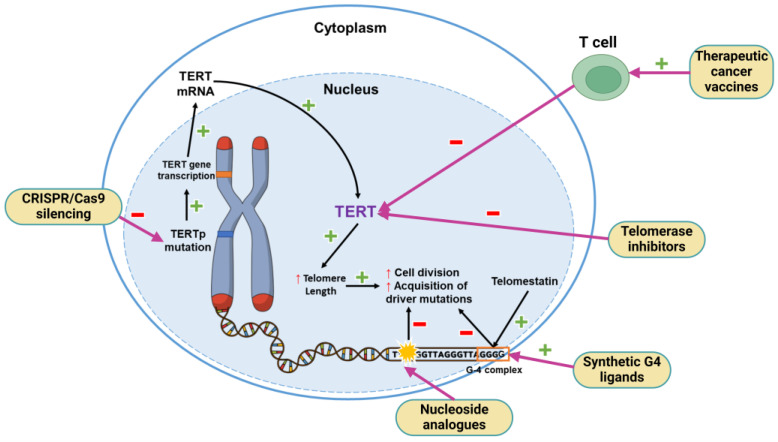
Representative mechanisms underlying telomere length maintenance via TERT promoter mutation in glioblastoma cells and targeted therapies against various sites in the pathway.

**Table 1 cancers-16-03635-t001:** Key features for glioblastoma diagnosis.

IDH-Wildtype and H3-Wildtype Positive Gliomas Plus Either:	
Histopathological features	Molecular features
-Microvascular proliferation -Necrosis	-TERT promoter mutation -EGFR amplification -Combined gain of chromosome 7 and loss of chromosome 10

**Table 2 cancers-16-03635-t002:** Selected therapies targeting TERT promoter currently under investigation.

Therapy	Mechanism of Action	Results	Ref.
**Imetelstat**	Telomerase inhibitor	Phase II clinical trial needed to be terminated due to two patients dying of intratumoral hemorrhage secondary to thrombocytopenia.	[103]
**INO-5401 + INO-9012 + cemiplimab**	Vaccine containing a DNA plasmid encoding human TERT, WT-1 and PSMA + a DNA plasmid encoding IL-12 + PD-1 inhibitor	Phase I/II study in newly diagnosed glioblastoma patients administered a DNA vaccine concomitantly with standard chemoradiation, demonstrating safety and a robust immune response and enhanced survival. Median OS in cohorts A (unmethylated MGMT) and B (methylated MGMT) was 17.9 and 32.5 months, respectively.	[105]
**UCPVax**	CD4 helper peptides derived from TERT	In a phase IIa trial in glioblastoma, IDH-wildtype, MGMT-unmethylated patients, the therapy was shown to be highly immunogenic and safe, improving OS.	[106]
**6-thio-2-deoxyguanosine (6-thio-dG)**	Nucleoside analogue that targets newly synthesized telomeres	This therapy caused telomere stress and DNA damage, whichactivated innate and adaptative immune-dependent responses. Not tested in glioblastoma cells.	[108]
**BIBR 1532 + ZIF-8**	Telomerase inhibitor in combination with zeolitic imidazole framework-8, a drug delivery vehicle	Reduced TERT mRNA expression with cell cycle arrest and cellular senescence with improved transportation and delivery of BIBR1532 to the nucleus. Not tested in glioblastoma cells.	[109]
**BRACO-19**	Synthetic G4 ligand	Decreased viability of human glioblastoma cells while sparing normal surrounding cells in a highly specific manner.	[110]
**Y2H2-6M(4)-oxazole telomestatin derivative (6OTD)**	Synthetic G4 ligand	Inhibition of the growth of GSCs and DNA damage, preferentially in telomeres of GSCs.	[111]
**RHPS4 + ionizing radiation**	Synthetic G4 ligand + ionizing radiation	Inhibition of growth in both differentiated glioblastoma and GSCs with synergistic radiosensitization in the differentiated cells but not in GSCs, possibly because effects on GSC growth inhibition are not mediated by telomeric dysfunction.	[112]
**Adenoviruses expressing sgRNA-guided CjABE**	sgRNA-guided and catalytically impaired Campylobacter jejuni CRISPR-associated protein 9-fused adenine base editor (CjABE) in an adeno-associated virus vector	Correction of TERTp mutation, reducing TERT transcription and TERT protein expression in human glioblastoma cell lines, which ultimately inhibited growth	[113]
**shRNA + temozolomide**	shRNAs targeting GABPB1L + alkylating agent chemotherapy	Reduction in the respective mRNA and protein levels, leading to reduced TERT mRNA and telomerase activity exclusively in TERTp-mutant glioblastoma cells. Chemotherapy sensitization resulted in increasing survival in a synergistic manner.	[114]

**Table 3 cancers-16-03635-t003:** Selected targeted EGFR therapies under investigation in adults with glioblastoma.

Therapy	Mechanism of Action	Results	Ref.
Erlotinib + Bevacizumab	First-generation EGFR inhibitor + monoclonal antibody against VEGF	Phase II clinical trial showed no improvement in OS in unmethylated MGMT glioblastoma patients.	[125]
Osimertinib + Bevacizumab	Third-generation EGFR inhibitor + monoclonal antibody against VEGF	Retrospective cohort study showed that the osimertinib/bevacizumab combination was marginally effective in most patients with simultaneous EGFR amplification plus EGFRvIII mutation, and a meaningful benefit was seen in a patient subgroup.	[126]
TAS2940	Irreversible pan-ErbB inhibitor	Improved brain penetration in in vitro and in vivo mouse xenograft models, inhibition of tumor growth against cells with HER2 amplification and EGFRvIII mutation, improving survival in mice. Ongoing clinical trials: NCT04982926, https://clinicaltrials.gov/study/NCT04982926, accessed on 27 October 2024.	[127]
Depatuxizumab mafodotin	Monoclonal antibody against EGFRvIII	Phase III clinical trial showed increased PFS but no improvement in OS versus placebo in newly diagnosed glioblastoma patients with EGFR. Corneal epitheliopathy occurred as an adverse effect in 94% of treated patients.	[128]
GC1118	Monoclonal antibody against EGFR	A multicenter, open-label, single-arm phase II trial demonstrated good tolerance and improved immune signatures in tumors but did not show benefit in terms of PFS or OS in patients with recurrent glioblastoma and EGFR amplification.	[129]
Panitumumab	Monoclonal antibody against EGFR	A cohort under the DRUP trial demonstrated safety of use but limited clinical benefit in only 21% of patients. Ongoing clinical trials: NCT03510208, https://clinicaltrials.gov/study/NCT03510208, accessed on 27 October 2024.	[130]
Nimotuzumab	Monoclonal antibody against extracellular region of EGFR	A phase II, single-arm, multicenter clinical trial showed increased OS in patients with newly diagnosed EGFR-expressed glioblastoma when added to standard chemoradiation. Importantly, MGMT status showed no correlation with these results.	[131]
Nimotuzumab + Melatonin	Monoclonal antibody against extracellular region of EGFR + hormone-blocking ATP binding to the kinase domain of EGFR	Synergistic increase in cytotoxicity and apoptosis of cancer cells in vitro and in xenograft mouse glioblastoma models.	[132]
Rindopepimut + Bevacizumab	Peptide-based vaccine targeting EGFRvIII + monoclonal antibody against VEGF	A double-blind, randomized phase II trial in recurrent EGFRvIII-expressing glioblastoma patients showed that concurrent administration of rindopepimut with Bevacizumab increased PFS at 6 months and OS compared to bevacizumab alone.	[133]
R-613	Oncolytic herpes simplex virus (oHSV) retargeted to EGFRvIII	The therapy delayed the development of tumor masses and increased OS in orthotopically transplanted mice when given as an early treatment.	[134]
OV-Cmab-CCL5	oHSV containing an IgG1 form of cetuximab and chemokine C-C motif ligand 5, a chemotactic chemokine	The therapy upregulated immune cell trafficking in the tumor microenvironment, with enhanced migration and activation of natural killer cells, macrophages and T cells; inhibition of tumor EGFR signaling; a subsequent reduction in tumor size; and increased survival in mouse models.	[135]
OV-IL15C	oHSV-expressing IL-15/IL-15Rα + CAR NK cells	Synergistic inhibition of tumor growth and increased survival in mouse models.	[136]
R-115	Fully virulent oHSV retargeted to human ErbB-2	A single injection showed significant improvement in the OS of mice and resistance to recurrence, with an unprecedented complete eradication of tumor in 30% of subjects.	[137]
CAR T-cell therapy	Chimeric antigen receptor T cells against EGFRvIII	A phase I trial demonstrated patient safety but no clinically significant change in PFS.	[138]
CAR T cells + Pembrolizumab	CAR T cells against EGFRvIII + monoclonal antibody against PD1	A phase I trial showed upregulation of the tumor microenvironment but no improvement in terms of PFS or OS.	[139]
Intrathecal CAR T cells	CAR T cells against EGFR and IL13Rα2	A phase I trial demonstrated safety, with only early-onset neurotoxicity and moderate efficacy and reductions in enhancement and tumor size detected by MRI in all patients.	[140]

**Table 4 cancers-16-03635-t004:** Selected targeted therapies against the Met signaling pathway.

Therapy	Mechanism of Action	Results	Ref.
Cabozantinib	TKI against VEGFR2 and MET	A multicenter, open-label, single-agent phase II trial enrolled recurrent glioblastoma patients, with a PFS at 6 months of 27.8% and an OS of 10.4 months, failing to meet the statistical target for success. Ongoing trials: NCT05039281 (https://clinicaltrials.gov/study/NCT05039281, accessed on 27 October 2024) (+ atezolizumab)	[156]
Capmatinib (INC280)	MET inhibitor	A multicenter, open-label, non-randomized, two-part study comprising a dose-escalation and expansion phase I trial including patients with various MET-positive solid tumors, including glioblastoma, showed that the drug was well tolerated and exhibited antitumor activity. Ongoing trials: NCT02386826 (https://clinicaltrials.gov/study/NCT02386826, accessed on 27 October 2024) (+ bevacizumab)	[157]
Crizotinib	ALK, *ROS1* and c-MET inhibitor	A multicenter, open-label, single-arm phase Ib trial demonstrated a median PFS of 10.7 months and OS of 22.6 months, showing a possible synergistic effect of crizotinib when was added to standard chemoradiation in newly diagnosed glioblastoma patients	[158]
Onartuzumab + bevacizumab	Monoclonal antibody against MET + monoclonal antibody against VEGF	A multicenter, randomized, double-blind, placebo-controlled phase II trial showed no clinical benefit of adding onartuzumab to bevacizumab when compared to placebo in recurrent glioblastoma patients.	[159]

**Table 5 cancers-16-03635-t005:** Selected targeted therapies for PTEN and regulation of the PI3K/Akt/mTOR signaling pathway.

Therapy	Mechanism of Action	Results	Ref.
Ad-PTEN + LY294002	Oncolytic adenovirus retargeted to upregulate PTEN + PI3K inhibitor	The combination of Ad-PTEN and LY294002 inhibited the PI3K/AKT pathway more effectively than either therapy alone, suppressing tumor growth in vitro and in in vivo glioblastoma xenograft mouse models.	[161]
Ipatasertib + atezolizumab	Akt inhibitor + monoclonal antibody against PD-L1	A single-center, open-label phase I/II trial showed the combination to be well tolerated, with clinical benefit in 32% of all patients and in 28.6% of patients with PTEN loss, making it a promising predictive biomarker for response to the combination.	[162,163]
Buparlisib (BK120)	PI3K inhibitor	A multicenter, open-label, multi-arm phase II trial in patients with recurrent glioblastoma showed significant brain penetration but incomplete blockade of the PI3K pathway and minimal efficacy.	[164]
Buparlisib + either carboplatin or lomustine	PI3K inhibitor + either platinum-based chemo or alkylating agent	A multicenter, open-label, randomized phase Ib/II trial in patients with recurrent glioblastoma showed insufficient antitumor activity compared with data on the use of single-agent carboplatin or lomustine.	[165]
Capmatinib + buparlisib	MET inhibitor + PI3K inhibitor	A multicenter, open-label phase Ib/II trial in patients with recurrent glioblastoma showed no clear activity using the combination.	[166]
Buparlisib + bevacizumab	PI3K inhibitor + monoclonal antibody against VEGF	A multicenter, phase I/II study in patients with recurrent glioblastoma showed poor tolerability of the combination, with 57% of patients experiencing at least one serious treatment-related toxicity and similar efficacy to that of single-agent bevacizumab.	[167]
Buparlisib + standard chemoradiation	PI3K inhibitor + alkylating agent + radiation therapy	A two-stage, multicenter, open-label phase I trial in newly diagnosed glioblastoma patients was not able to determine the maximum tolerated dose due to dose-limiting toxicities. Subsequently, Novartis decided not to pursue the development of buparlisib in newly diagnosed glioblastoma.	[168]
Paxalisib + metformin	PI3K/mTOR inhibitor + biguanide	The therapy increased the efficacy of PI3K inhibitors when combined with metformin and a ketogenic diet to reduce insulin feedback and hyperglycemia in orthotopic glioblastoma mouse models. Ongoing trials: NCT05183204 (https://clinicaltrials.gov/study/NCT05183204, accessed on 27 October 2024)	[169]
AZD-9291 + GDC-0084	EGFR/MEK/ERK pathway inhibitor + PI3K/AKT/mTOR inhibitor	Synergistic inhibition of proliferation and survival in in vitro and in vivo glioblastoma mice models was seen with this combination.	[170]

**Table 6 cancers-16-03635-t006:** Selected therapies targeting MGMT protein activation.

Therapy	Mechanism of Action	Results	Ref.
d3A + temozolomide	Chimeric fusion protein (CRISPR/dCas9 + DNA methyltransferase 3A catalytic domain) + alkylating agent	Targeted MGMT methylation in specific CpG clusters in the vicinity of the promoter, with consequent MGMT downregulation and enhanced chemosensitivity to temozolomide of glioblastoma cells in vitro.	[176,177]
Folic acid + temozolomide	Water-soluble vitamin + alkylating agent	A phase I trial demonstrated the safety of adding folic acid and restored methylation of the promoter in samples of circulating tumor DNA of 8 glioblastoma patients.	[178]
L-methylfolate + temozolomide + bevacizumab	Water-soluble vitamin + alkylating agent + monoclonal antibody against VEGF	A phase I trial showed the safety of L-methylfolate and DNA methylome reprogramming of recurrent IDH-wild-type glioblastomas. However, no significant impact on survival could be demonstrated due to a lack of statistical power.	[179]
Lomeguatrib	MGMT inhibitor	Inactivation of MGMT protein in glioblastoma cells in vitro, with increased radiosensitization at lower concentrations and radioprotective effects at higher doses.	[180]
5-Azacitidine (5-Aza) + lomeguatrib	Demethylating agent + MGMT inhibitor	Epigenetic reactivation of TUSC3, which increased GSC sensitivity to temozolomide in MGMT-unmethylated orthotopic GSC mouse models.	[181]
Bortezomib + temozolomide	26S proteasome inhibitor + alkylating agent	Depletion of MGMT mRNA and protein in glioblastoma cells in vitro and diminished proteasome activity in orthotopic mouse models, with increased survival. Ongoing trials: NCT03643549 (https://clinicaltrials.gov/study/NCT03643549, accessed on 27 October 2024)	[182]
Marizomib + standard treatment	Pan-proteasome inhibitor ± alkylating agent and radiotherapy	A multicenter, randomized, controlled, open-label phase III trial in patients with newly diagnosed glioblastoma showed increased toxicity and no statistically significant difference in OS or PFS between patients receiving marizomib in addition to standard treatment (RT/Temozolomide) compared with patients receiving standard treatment alone, either in the MGMT-methylated or unmethylated subgroup.	[183]

## Data Availability

Not applicable.

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
