# Peer review of "Evolution of Molecular Biomarkers and Precision Molecular Therapeutic Strategies in Glioblastoma"

_cancers, 2024, doi:10.3390/cancers16213635_

Round 1
Reviewer 1 Report
Comments and Suggestions for Authors
In this review article titled 'Evolution of Molecular Biomarkers and Precision Molecular 2 Therapeutic Strategies in Glioblastoma' Jacome et al have reviewed histopathology, radio-genomics and targeted therapies for glioblastoma.
I don't have major concerns with the review, just a few minor points:
1) line 90 and the lines in this section: the authors don't really shed much light on the debate about the cells of origin in GBM. They only discuss the cells from the SVZ. Authors should also mention the other cells of origin.
2) Section 5.1, would benefit from adding a few sentences about Voraseidenib, the new drug approved last year for IDH mutant low grade gliomas.
3) Section 5.6. NTRK fusion proteins have been characterised in Glioma by Patwell et al in 2020. (https://pubmed.ncbi.nlm.nih.gov/32532995/)
Typos or words to be checked:
Line 77- NCSs should be NSCs
Table 1 -molecular features underlined and separated from histological features
Line 108 heterogenecity?
Line 130 with series? what does it mean?
table3: intratecal CART
Comments on the Quality of English Language
English language can be improved
Author Response
Response to Reviewer 1
Opening comment: “In this review article titled 'Evolution of Molecular Biomarkers and Precision Molecular 2 Therapeutic Strategies in Glioblastoma' Jacome et al have reviewed histopathology, radio-genomics and targeted therapies for glioblastoma. I don't have major concerns with the review, just a few minor points:
Response: Thank you kindly for taking the time to read and comment our manuscript.
Minor comment 1: line 90 and the lines in this section: the authors don't really shed much light on the debate about the cells of origin in GBM. They only discuss the cells from the SVZ. Authors should also mention the other cells of origin.
Response: We appreciate the reviewer for pointing this out. We acknowledge the wording of this section was confusing. Following the recommendation, we revised the text, rearranged some paragraphs and added a sentence stating the other cells listed along with NSCs are also considered cells of origin for glioblastoma, also emphasizing the debate has not reached any conclusion. We hope is now clearer. Please see page 2 of the revised manuscript, lines 81-88.
Minor comment 2: Section 5.1, would benefit from adding a few sentences about Vorasidenib, the new drug approved last year for IDH mutant low grade gliomas.
Response: We appreciate the suggestion and added a couple of sentences about Vorasidenib as it emphasizes that targeting mutant IDH has proved feasible. When contrasting this with strategies to target IDH wildtype we think it helps to stress the gap in research and therapies between low grade gliomas and glioblastomas. The lines added are in page 7 of the revised manuscript, lines 326-329.
Minor comment 3: Section 5.6. NTRK fusion proteins have been characterized in Glioma by Patwell et al in 2020. (https://pubmed.ncbi.nlm.nih.gov/32532995/)
Response: We thank the reviewer for presenting us with this article. We found it very interesting and on theme with our review. We have incorporated a few lines (see page 20, lines 765-770) to discuss the findings about NRTK fusion and splice variants and their perspective on how this affect TRK inhibitors resistance, which is pointed in a posterior section as a major gap in literature of targeted therapies for glioblastoma.
Typos or words to be checked:
Typo 1: Line 77- NCSs should be NSCs
Response: We’ve corrected the typo to “NSCs”
Typo 2: Table 1 -molecular features underlined and separated from histological features
Response: We’ve corrected the format of the titles and are now both underlined over their respective lists of features
Typo 3: Line 108 heterogenecity?
Response: We’ve corrected the typo to “heterogeneity”
Typo 4: Line 130 with series? what does it mean?
Response: We apologize for this confusion. We have decided to rephrase “with some series showing positivity in more than 90% of glioblastomas” to “with some authors finding at least one positive marker in more than 90% of glioblastomas”. We hope this makes it clearer.
Typo 5: table3: intratecal CART
Response: We’ve corrected the typo to “intrathecal”

Reviewer 2 Report
Comments and Suggestions for Authors
1) Synopsis
The paper titled "Evolution of Molecular Biomarkers and Precision Molecular Therapeutic Strategies in Glioblastoma" reviews the latest advances in molecular biomarkers and targeted therapies for glioblastoma. It discusses the integration of histopathological and molecular findings in glioblastoma diagnosis, the evolution of molecular markers such as IDH mutation, TERT promoter mutation, and EGFR amplification, and the development of novel therapeutic strategies, including precision medicine approaches. The review highlights advancements in radiogenomics, immunotherapies, and molecular-targeted therapies, aiming to refine glioblastoma treatment protocols and improve patient outcomes.
2) Strengths
- Comprehensive Overview: The paper offers a detailed and structured review of the molecular biomarkers involved in glioblastoma, supported by recent clinical studies and research trials.
- Clinical Relevance: The discussion on novel therapies, particularly in precision medicine and immunotherapy, reflects the importance of these approaches in glioblastoma treatment.
- Multidisciplinary Approach: It integrates molecular biology, clinical trials, and radiological features, providing a holistic understanding of glioblastoma from a research and clinical perspective.
3) Weaknesses
- Lack of Novelty: The paper mostly summarizes existing studies without introducing new perspectives or innovative insights into the field. Much of the content is a reiteration of known molecular pathways and current therapeutic options, limiting its novelty in the context of recent literature.
- Limited Critical Evaluation: There is insufficient critical analysis of the limitations and challenges associated with the biomarkers and therapies discussed. The review would benefit from a deeper exploration of the shortcomings in current clinical applications and future directions.
- Schematic Representations: The paper lacks innovative visual aids (e.g., diagrams or flowcharts) to simplify complex molecular interactions and therapeutic strategies, which could enhance the reader's understanding.
Revision Requests
- Highlight the Novelty Deficit: The paper lacks innovative insights beyond summarizing existing data. The authors should differentiate their review by incorporating novel ideas, emerging trends, or gaps in research that are not widely discussed in the current literature.
- Critical Evaluation of Literature: Add a section dedicated to critically assessing the existing literature, specifically pointing out the limitations of current molecular markers and therapies. Discuss where clinical trials have failed or shown conflicting results, providing a balanced view of progress and hurdles.
- Innovative Schematics: Include detailed schematics or diagrams to represent molecular pathways, biomarker interactions, and therapeutic mechanisms in glioblastoma. These visual aids can illustrate the complexity of glioblastoma biology and how targeted therapies intervene.
- Comparison to Online Available Resources: Compare your review with recent meta-analyses and databases available online to show where your paper adds value. Identify key research gaps that are underrepresented in current resources and explain how your review addresses them.
- Detailed Case Studies: Provide examples or case studies from recent clinical trials to substantiate your review of therapeutic strategies. Highlight where certain biomarkers or treatments have shown promise in individual patients and why these cases are clinically significant.
- Incorporate AI/ML in Radiogenomics: Discuss how artificial intelligence (AI) and machine learning (ML) are being integrated into radiogenomics for glioblastoma. This is an emerging field that could make the review more forward-thinking and aligned with future advancements in diagnosis and treatment.
- Expand on Treatment Resistance Mechanisms: Discuss the challenges and mechanisms of resistance to therapies, particularly in glioblastoma's molecular heterogeneity. Including this will make the review more critical and less descriptive.
- Include Future Perspectives Section: Add a section forecasting the future of glioblastoma research, focusing on areas like nanomedicine, personalized treatment models, and advances in multi-omic integration for precision oncology.
Author Response
Response to Reviewer 2
Opening comments:
1) Synopsis
The paper titled "Evolution of Molecular Biomarkers and Precision Molecular Therapeutic Strategies in Glioblastoma" reviews the latest advances in molecular biomarkers and targeted therapies for glioblastoma. It discusses the integration of histopathological and molecular findings in glioblastoma diagnosis, the evolution of molecular markers such as IDH mutation, TERT promoter mutation, and EGFR amplification, and the development of novel therapeutic strategies, including precision medicine approaches. The review highlights advancements in radiogenomics, immunotherapies, and molecular-targeted therapies, aiming to refine glioblastoma treatment protocols and improve patient outcomes.
2) Strengths
- Comprehensive Overview: The paper offers a detailed and structured review of the molecular biomarkers involved in glioblastoma, supported by recent clinical studies and research trials.
- Clinical Relevance: The discussion on novel therapies, particularly in precision medicine and immunotherapy, reflects the importance of these approaches in glioblastoma treatment.
- Multidisciplinary Approach: It integrates molecular biology, clinical trials, and radiological features, providing a holistic understanding of glioblastoma from a research and clinical perspective.
3) Weaknesses
- Lack of Novelty: The paper mostly summarizes existing studies without introducing new perspectives or innovative insights into the field. Much of the content is a reiteration of known molecular pathways and current therapeutic options, limiting its novelty in the context of recent literature.
- Limited Critical Evaluation: There is insufficient critical analysis of the limitations and challenges associated with the biomarkers and therapies discussed. The review would benefit from a deeper exploration of the shortcomings in current clinical applications and future directions.
- Schematic Representations: The paper lacks innovative visual aids (e.g., diagrams or flowcharts) to simplify complex molecular interactions and therapeutic strategies, which could enhance the reader's understanding.
Response: We appreciate the reviewer taking the time to write such an insightful and detailed review. We found your comments very constructive and perceptive. We hope to have improved some of the weaknesses addressed and provide a revised version that is valuable.
Revision Request comments:
Comment 1: Highlight the Novelty Deficit: The paper lacks innovative insights beyond summarizing existing data. The authors should differentiate their review by incorporating novel ideas, emerging trends, or gaps in research that are not widely discussed in the current literature.
Response: We appreciate the reviewer for pointing this out and we agree that a discussion on the flaws and limitations this paper has can serve as a reminder that no study is perfect and there is always room for improvement and new approaches for investigation. We added several paragraphs to expose the areas in which this study could be complemented, including some of the ones suggested by the reviewer. We hope this provides readers with an idea of where future investigation can be guided. These can be found in page 21-22, “Discussion and Future Perspectives” section.
Comment 2: Critical Evaluation of Literature: Add a section dedicated to critically assessing the existing literature, specifically pointing out the limitations of current molecular markers and therapies. Discuss where clinical trials have failed or shown conflicting results, providing a balanced view of progress and hurdles.
Response: As a Literature Review, we aimed to comprehensively summarize the latest evidence on molecular markers and targeted therapies of glioblastoma, providing an overview of the trends that have been explored in the last five years. In each subsection of section 5 we explored some of the conflicting results, some approaches have shown throughout the years, listing failures and conflicting results among them. This is also summarized in tables 2-6. Meanwhile, we think adding a full section to specifically assess limitations of these markers and therapies can be redundant and beyond the scope of our paper and comparing results between clinical trials can be better addressed as a Systematic Review in our future study.
Comment 3: Innovative Schematics: Include detailed schematics or diagrams to represent molecular pathways, biomarker interactions, and therapeutic mechanisms in glioblastoma. These visual aids can illustrate the complexity of glioblastoma biology and how targeted therapies intervene.
Response: We appreciate this observation and we agree that incorporating visual aids on molecular pathways and how targeted therapies interact with them will enhance comprehension for readers and make the review more memorable. We’ve created 3 figure schematics: Figure 1 (page 7) briefly summarizes all molecular markers and therapies discussed in this review. Figure 2 (page 8) illustrates IDH and MGMT cell mechanisms and action targets for those markers in glioblastoma. Figure 3 (page 11) illustrates telomere metabolism in glioblastoma and different targeted therapies.
Comment 4: Comparison to Online Available Resources: Compare your review with recent meta-analyses and databases available online to show where your paper adds value. Identify key research gaps that are underrepresented in current resources and explain how your review addresses them.
Response: We appreciate the insightful suggestion. We have found that meta-analysis and other reviews on targeted therapies in glioblastoma usually focus on only one or a few markers at a time. With our review we tried to be as comprehensive as possible and organize the data in a way that could potentially be used as a guide to determine the direction for further analysis such as systematic reviews or meta-analysis. We have conveyed this in the Conclusions sections in page 22.
Comment 5: Detailed Case Studies: Provide examples or case studies from recent clinical trials to substantiate your review of therapeutic strategies. Highlight where certain biomarkers or treatments have shown promise in individual patients and why these cases are clinically significant.
Response: Thank you for the comments. We agree with the reviewer that individual examples in which a patient has benefited from a therapeutic strategy can help making a case for certain therapies showing conflicting results. Along the subsections on targeted therapies, we discussed clinical trials. We feel that adding punctual cases won’t change the overall sense of the review and might reiterate what’s already been conveyed when discussing results from clinical trials.
Comment 6: Incorporate AI/ML in Radiogenomics: Discuss how artificial intelligence (AI) and machine learning (ML) are being integrated into radiogenomics for glioblastoma. This is an emerging field that could make the review more forward-thinking and aligned with future advancements in diagnosis and treatment.
Response: Thank you for this great suggestion. We agree that this is an emerging field that will influence all future advancement. In section 4.3. Radiogenomics in tumors of the brain, we already discussed some of the ways in which AI and ML are being used in radiogenomics for glioblastoma and other brain tumors. Following the suggestion of the reviewer, we added a few lines on this in section 6 (see page lines 779-786). This section has been renamed “Discussion and Future perspectives” to also comply with comment #8.
Comment 7: Expand on Treatment Resistance Mechanisms: Discuss the challenges and mechanisms of resistance to therapies, particularly in glioblastoma's molecular heterogeneity. Including this will make the review more critical and less descriptive.
Response: We thank the reviewer for the comments. We agree with the reviewer that further elaborating on this point can enhance the final scope the review provides. In section 6 we discussed how understanding some resistance mechanisms has been difficult and represented a gap in scientific literature. We also added a few lines referring to this in section 7: Conclusions.
Comment 8: Include Future Perspectives Section: Add a section forecasting the future of glioblastoma research, focusing on areas like nanomedicine, personalized treatment models, and advances in multi-omic integration for precision oncology.
Response: Although we had already included a Future Perspectives section, this insightful observation made us reevaluate it and make some changes to improve it. Firstly, we changed the title of section 6 from “Future Directions” to “Discussion and Future Perspectives” as this modified version discusses what’s addressed in the review and offers our perspective on what potentially could be the future of investigation for glioblastoma. We also added some lines about nanomedicine in glioblastoma, including a mention to a recently published systematic review. We mentioned personalized medicine as the reason to continue investigation on targeted therapies. Along with radiogenomics, we discussed the need for integrating multi-omics analysis in other areas of investigation such as resistance mechanisms. We hope this revised version can provide a more critical perspective of the review.

Round 2
Reviewer 2 Report
Comments and Suggestions for Authors
I admire careful revision by the respected author. I agree with their logical arguement on some of my comments.